# Research on Seasonal Thermal Neutral Temperature in West Lake Scenic Area of Hangzhou, China

**DOI:** 10.3390/ijerph192214677

**Published:** 2022-11-09

**Authors:** Yi Mei, Lili Xu

**Affiliations:** School of Design and Architecture, Zhejiang University of Technology, Hangzhou 310027, China

**Keywords:** thermal evaluation methods, thermal neutral temperature (range), thermal acceptable temperature range, youth group, seasonality

## Abstract

The thermal neutral temperature is the foundation for the evaluation of the outdoor thermal environment sensation. This study is designed to investigate the thermal neutral temperature of the outdoor space of Hangzhou West Lake. Both the median method and the thermal sensation vote (TSV) = 0 methods are adopted to discuss the seasonal thermal neutral temperature, thermal neutral temperature range, and thermal acceptable temperature range of the youth group with college students in Hangzhou as representatives. Via the analysis of the relationship between TSV and thermal evaluation index PET, the median method, which proved more suitable for the study site, is selected to obtain the thermal evaluation results. It’s found from the study that, throughout the year, the thermal neutral temperature of Hangzhou West Lake scenic area is 21.0 °C; the thermal neutral temperature range is 12.0–23.9 °C, and the acceptable thermal temperature is 13.0–25.7 °C. The youth group visiting Hangzhou West Lake has the highest acceptance of the thermal environment in spring and autumn and the lowest in winter. Furthermore, the empirical models show that air temperature and average wind speed are the key factors impacting the thermal evaluation of the youth group. This study can serve as a reference for thermal evaluation in similar climate regions.

## 1. Introduction

The outdoor climate environment is an important factor impacting the activities of urban people. A pleasant climate can extend people’s activity time, improve social efficiency and behavioral intensity in outdoor space, as well as improve people’s quality of life, happiness index, and health level. The thermal nature temperature in thermal environment evaluation is an important indicator that impacts human outdoor thermal perception. In terms of the thermal nature of temperature, attention should be paid to the investigation of individual factors (such as individual health, tolerance, and adaptability) [1], and it is defined as the temperature at which the centered thermal sensation occurs in a linear regression between Mean thermal sensation vote (MTSV) and thermal sensation temperature [2]. Based on the thermal nature of temperature, it has been further proposed that a certain interval threshold instead of a fixed value should be used to assess the individual’s acceptance of thermal conditions, thus enhancing the effectiveness of the evaluation model in responding to real human feelings [3]. The thermally acceptable temperature range is the thermal evaluation index derived from this. This evaluation index starts from the different adaptability of individuals to the thermal environment and calculates the temperature perception thresholds within a certain range generated by the fluctuating influence of various factors, such as the state of the thermal environment and the duration of the thermal stimulus in humans [4]. Most of the current thermal evaluation studies are mainly conducted in the form of reports such as questionnaires or interviews, which report results divided by odd scales, graded individual perceptions, and qualitative analysis of human thermal comfort [5]. Quantitative methods such as multiple linear regression analysis, linear transformation, and analysis of variance for different ages, gender, and residence populations have also been used [6] to explore the human evaluation of thermal environments. In the existing studies on the thermal neutral temperature or thermal center temperature range in different climate regions, there are two corresponding ways to obtain the thermal neutral temperature or thermal neutral temperature range. One is the imputation of thermal nature temperature (range) based on the subjective polling of respondents via the use of the median MTSV method [7,8,9]; the other is the calculation of thermal nature temperature and thermal nature temperature range within the threshold of TSV = 0 or −0.5 ≤ MTSV ≤ 0.5 via the use of the linear regression equation of TSV and PET [10]. A comprehensive comparison of various types of relevant data [11,12] reveals that design studies that respect regional characteristics have increasingly emphasized that the selection of research methods should be optimized according to the climatic conditions of specific regions. However, studies that highlight the applicability of research methods in specific regions and emphasize the possible bias of conclusions due to method selection are less likely to be conducted. Therefore, in this study, the most suitable method among the above two research methods for the West Lake scenic area in Hangzhou is selected based on the measured data. Further multiple regression analysis is adopted to calculate the thermal nature temperature and thermal nature temperature range in the area.

The literature on the thermal comfort of urban waterfront tourist attractions is still relatively rare. The existing inter-seasonal comparative studies mostly focus on the urban settlement environment. The research object groups are mostly local residents, and the thermal comfort is basically at a stable level. There are obvious differences between the Figscenic tourist areas and the urban settlements when talking about the perception of thermal comfort. Hangzhou West Lake Scenic Area is an internationally famous tourist attraction, attracting a large number of domestic and foreign tourists. The upcoming Asian Games in 2023 will usher in more visitors to Hangzhou West Lake. Compared with local residents, these tourists from different regions are more likely to have significantly different physiological responses to the thermal experience during a short trip. The social application value of this study is mainly reflected in the fact that it can provide a clearer choice for tourists with different needs in different seasons to better enhance the tourism experience. That is why seasonal thermal neutral temperature research in West Lake should be conducted. In this study, the youth group of college students is selected as the research target. The following objectives will be achieved through the analysis of the measured data:(1)to clarify the thermal evaluation method applicable to the West Lake of Hangzhou.(2)to evaluate the seasonal differences in thermal perception of the recreational space of the West Lake of Hangzhou, and furthermore to derive the thermal nature temperature, thermal nature temperature range, and thermally acceptable temperature range by season.(3)to examine the meteorological variables influencing the thermal perception of Hangzhou’s West Lake and to suggest an empirical model for thermal perception prediction and its seasonal variations.(4)to contribute to the design of landscape microclimate adaptability and crowd tour route.

## 2. Study Ideas and Methods

### 2.1. Indicator Selection

There are dozens of applied thermal evaluation indicators, among which Physiological Equivalent Temperature (PET) [13] is a thermal indicator developed by Höppe from the Munich energy balance model for individuals (MEMI- model). It is defined as the air temperature of the thermal environment in an indoor environment when the core and skin temperatures are the same as in the actual environment, and the same thermal equilibrium is maintained. The PET index has been developed to have the advantage of being able to accurately and intuitively reflect the real heat stress and thermal comfort of the user under more complex integrated outdoor thermal conditions [14]. The advantages of this index also include (1) its units (°C) can be easily understood and applied by urban planners, decision-makers, and users; (2) the strong correlation between PET and Thermal Sensation Vote (TSV) is conducive to a combined subjective and objective comparative evaluation. Therefore, PET is used as a thermal indicator to predict and evaluate outdoor thermal perception. Based on the team’s previous research results [15], the selected key outdoor climate environment indicators include air temperature, solar radiation, average wind speed, and relative air humidity.

### 2.2. Experiment and Methods

#### 2.2.1. Actual Measurement Time

In this study, Hangzhou urban meteorological data for the past 30 years (Figure 1) are counted, and the results show that the summer season in Hangzhou generally starts in early June and ends in early October each year, occupying about half one year, and the other three seasons last 2–3 months, respectively. However, it’s found from preliminary experiments that the rainy season in late spring and the frequent typhoons in early autumn led to a significant decrease in the number of outdoor recreations during this period, leading to the failure to meet the experimental sample size requirement. Therefore, based on the meteorological phenology method (i.e., the season in which the average temperature is over 22 °C in a unit time of 5 consecutive days is considered as summer, and the season in which the average temperature below 10 °C is winter. The transition season from winter to summer is spring, and the transition season from summer to winter is autumn.) Based on the method of dividing the four seasons, the spring and autumn seasons are jointly considered transitional seasons, and the statistics are combined. Typical meteorological days (Table 1) are selected for each season. The experiment is conducted from 8:00 to 18:00 daily, with the experiment not conducted in winter 2021 due to the impact of COVID-19.

#### 2.2.2. Measurement Point Layout

In the study, the types of open space, daily usage rates, and meteorological factors are compared in the West Lake scenic area of Hangzhou, and four typical scenic garden spaces in the West Lake loop are selected as experimental measurement points, with one test instrument and many measurement personnel at each measurement point. Four measurement points are, respectively, Lakeside Sunny and Rainy in the northeast corner of the West Lake Lakeside, the Long Bridge Park in the southeast corner, Wangshan Bridge in the southwest corner, and Autumn Moon over the Calm Lake in the northwest corner. All four points are equipped with a waterfront space and a small square where people can gather. The layout of the measurement points and the experimental photos are shown in Figure 2, which is repainted from Baidu Map.

#### 2.2.3. Actual Measurement Instrument

In the experiment, The YGY-QXY handheld meteorological instrument is used to measure meteorological indicators such as air temperature, solar radiation, average wind speed, and relative air humidity. The measurement range, accuracy, and data collection frequency of the instrument are in line with relevant regulations in ISO 7726:1998, and the measurement interval is set as 15 min.

#### 2.2.4. Questionnaire Design

The questionnaire is designed based on ISO 10551:1995 Thermal Ergonomics and GB/T 18977-2003. Here, the odd-numbered voting method is adopted to assess the effects of the thermal environment on the human body. The questionnaire contains (1) basic information about the respondents, including gender, age and length of time in the hang, etc.; (2) respondents’ heat sensation voting and heat acceptability voting. Subjective questionnaires and information collection are conducted in parallel with objective meteorological data collection.

## 3. Results

To make sure there are sufficient samples of the same kind, the respondents are all university students in school. Although the respondents are from different regions, they have stayed in Hangzhou for over two years and have already adapted to Hangzhou’s climatic conditions, so the interference factors brought by their original residence can be excluded. A total of 400 questionnaires are distributed. Of the 400, 387 are returned, of which 360 are valid, including 155 in the transitional season, 118 in the summer, and 87 in the winter. Due to the climatic environment and COVID-19 in winter, long-term experiments can’t be conducted, so the number of those counted is lower than that in the other two seasons. The Cronbach α coefficient of the combined reliability of the questionnaire data is 0.813. Here the coefficient α > 0.8, indicating the high quality of data reliability, which can be used for further analysis.

### 3.1. Meteorological Data Measurement Results

The maximum values of each meteorological index collected by season (Table 2) are 1037 W/m^2^ of solar radiation in the transitional season, 43.6 °C of air temperature in summer, 72.3% of relative air humidity, and 4.4 m/s of average wind speed in winter. The maximum daily difference in solar radiation is 1063.0 W/m^2^ in summer, and the minimum is 558.0 W/m^2^ in winter; the maximum daily difference in temperature is 24.8 °C in the transitional season, which is 7 °C higher than that in the summer; the maximum daily difference in mean relative air humidity is 34.5% in the summer, and the minimum difference is 23.5% in the transitional season; the maximum daily difference in mean wind speed is 4.4 m/s in the winter, and the minimum is 2.2 m/s in the summer.

### 3.2. Thermal Sensation Vote

The thermal sensation was used in the study of the Built Environment for the first time. The conclusion of thermal sensation in this study was derived from the thermal sensation vote (TSV). The thermal sensation, which is closely related to the personal experience can be influenced by the environment but cannot be precisely measured or described [16,17]. Thus, the thermal sensation of subjects could only be shown in the form of questionnaires with the content of a seven-level index consistent, which is the ASHRAE thermal sensation scale (−3~+3). The result of the questionnaire is shown in Figure 3. Although the highest number of TSV votes in each season is concentrated in the “medium” option, the results still show distinct seasonal differences. Transitional season voting shows a tendency to be cooler, with 81.31% of votes falling in the “medium,” “slightly cool,” and “cool” categories; summer voting is concentrated in the “medium,” “cool,” and “cool” categories. In summer, the votes are concentrated in “medium,” “slightly warm,” and “warm,” with a total of 82.81% of the votes; in winter, 91.78% of the votes are concentrated in the “slightly cool,” “medium” and “warm” categories. Since TSV is a subjective vote, which is easily influenced by immediate psychological changes and individual experiences, the seasonal differences in the voting results are obvious.

### 3.3. Thermal Preference Vote

The seasonal preference voting statistics for air temperature, solar radiation, average wind speed, and relative humidity indicators are divided into three categories: “weaker”, “invariant”, and “stronger”, and the results are shown in Figure 4. Respondents’ preference for relative humidity is most obvious in the transitional season, with about half of them considering that the humidity is too high; the subjective preference for air temperature and average wind speed is stronger in summer, with most of them considering summer temperature to be too high and hoping for greater wind speed; in winter, respondents focus on solar radiation and average wind speed, with a relative preference for stronger sunshine and lower wind speed. From Figure 4, it can be seen that: the main meteorological indicators impacting the transitional season are concentrated on average wind speed and relative air humidity, while in summer, they are concentrated on air temperature and solar radiation, and in winter, they are concentrated on air temperature and relative air humidity. The sum of the votes for “invariant” among the indicators in the transitional season is the highest, while that of the votes for this option in summer and winter is relatively low. It is tentatively assumed that the respondents have the best thermal evaluation of the transitional season.

## 4. Discussion and Analysis

### 4.1. Thermal Neutral Temperature (Range) vs. Thermal Acceptable Temperature Range

Consistent with the findings of thermal comfort studies in other regions [18], the experimental results of this study demonstrate a strong correlation between TSV and PET, thereby confirming that the PET index can be better used to predict the thermal evaluation of the West Lake scenic area. In the following section, the relationship between these two is compared, and the more suitable method among the median method and TSV = 0 method for the follow-up study of the West Lake scenic area in Hangzhou is selected.

#### 4.1.1. Median Method

Based on the property that the median is not affected by interfering data and can accurately reflect the main data, box line plots are drawn for each season with TSV as the category axis and PET as the variable (Figure 5). The upper quartile and lower quartile of PET values in each season in the statistical experimental data are calculated. The threshold value located between the two quartiles is regarded as the thermally acceptable range, and the resulting thermally acceptable temperature ranges are 10.8–25.7 °C in the transitional season, 17.4–26.4 °C in the summer, 7.1–24.4 °C in the winter, and 13.0–25.7 °C throughout the year. The temperatures at the neutral point are considered thermally neutral and are 22.2 °C in the transitional season, 23.6 °C in the summer, 12.0 °C in the winter, and 21.0 °C throughout the year. According to ASHRAE Standard 55 [19], the acceptable thermal condition, i.e., the thermally acceptable temperature range, should be acceptable to at least 90% of the respondents in the same space. That means <10% of the users are allowed not to accept the thermal condition. According to this criterion, 3.6%, 5.9%, and 9.2% of the respondents maintain an “unacceptable” attitude towards the climate conditions of the transitional season, summer, and winter, respectively. Among them, the percentage of thermal unacceptability is highest in winter and lowest in the transitional season. The acceptable temperature range is10.8–25.7 °C in the transitional season, 17.4–26.4 °C in the summer, 7.1–24.4 °C in the winter, and 13.0–25.7 °C throughout the year.

#### 4.1.2. TSV = 0 Method

In the TSV = 0 method, the mean thermal sensation vote MTSV is adopted as the calculation index. MTSV represents the mean TSV within a 1 °C PET interval (PET = ±0.5 °C). MTSV = 0 is substituted into the linear regression model to calculate the thermal nature temperature (Figure 6), and then the relationship between PET and MSTV is discussed. The positive correlation between PET and MTSV is described. The ASHRAE standard specifies that all types of thermal sensation values should be within ±0.5, so the thermal nature temperature range is calculated within the −0.5 ≤ MTSV ≤ 0.5 thresholds.

The formula for each season of the whole year is shown as follows:

Transitional season: MTSV = 0.1118 PET − 1.6786 (R^2^ = 0.714, *p* < 0.01).

Summer: MTSV = 0.1596 PET − 3.2371 (R^2^ = 0.842, *p* < 0.01).

Winter: MTSV = 0.1378 PET − 2.3387 (R^2^ = 0.723, *p* < 0.01).

Annual: MTSV = 0.1249 PET − 2.1588 (R^2^ = 0.754, *p* < 0.01).

The thermally acceptable range is delineated as the −1 ≤ MTSV ≤ 1 threshold range, which means that MTSV < −1 or TSV > 1 is considered the thermally unacceptable range. The thermally neutral temperatures calculated by MTSV = 0 are 15.0 °C in the transitional season, 20.3 °C in the summer, 17.0 °C in the winter, and 17.3 °C throughout the year. The thermally acceptable temperature ranges calculated based on the above formula are 6.1 to 24.0 °C in the transitional season, 14.1 to 26.5 °C in the summer, 9.7 to 24.2 °C in the winter, and 9.3 to 25.3 °C in throughout the year.

### 4.2. Result from Comparison and Selection

The median method and the TSV = 0 methods are used to derive the thermally neutral temperature (range) and the thermally acceptable temperature range in Hangzhou, respectively. The results are shown in Table 3. It can be seen from Table 3 that there are significant differences between the two methods. Here, the median method yields relatively stable seasonal variations and small variations, and the TSV = 0 method yields large seasonal fluctuations in thermal neutral temperature, with low thermal neutral temperature in the transitional season, summer, and year-round, and high thermal neutral temperature in winter. The thermal nature temperature (range) obtained by the above two methods is further compared with the PET temperature (range) calculated from the measured meteorological data. It’s found that the median method is closer to the PET value, i.e., it’s more consistent with the actual meteorological and environmental conditions of the West Lake scenic area in Hangzhou. Therefore, the median method is chosen as the research method, and the subsequent discussion is made based on the experimental data of the median method.

### 4.3. Comparison of Thermal Sensations

In terms of seasonal differences in thermally acceptable temperatures, the study data indicate that Hangzhou college students are more adapted to mild climatic conditions in the transitional seasons. In the remaining two seasons, they are more likely to tolerate the hot summer than the cold winter. This is similar to the findings of studies on perceived changes in outdoor temperature in Guangzhou [3], Changsha [20], Shanghai campus [21], and Biskra [22] in hot and humid regions and Tehran [2] and Beijing [23] in dry regions. It can be found from these experiments that subjects are more adapted to warm and hot climatic conditions. However, some scholars hold a different view that the subject populations in cold regions such as Tianjin [24] and Xi’an [25] are more adapted to colder conditions. Although there is some agreement difference in the acceptance of extreme climatic conditions in the winter and summer seasons, scholars agree that outdoor space users are less thermally acceptable in summer and winter than in the transitional season. In other words, the climate in the transitional season is the most comfortable. It confirms the speculation in the previous section. The final results of this paper show that the transitional season has the highest number of voters for the optimal thermal comfort state; the thermally neutral temperature in the transitional season is 22.2 °C, just followed by that in summer −23.6 °C, then by that in winter −12.0 °C. The conclusion of this study can support the design strategy that in the climate adaptation design of Hangzhou West Lake scenic area, priority should be given to winter and summer, then to the transitional season when the design practice is guided.

Regarding the age differences of the experimental subjects, the thermal nature temperatures obtained in this study for the youth population are overall low compared to those obtained by Xiaoshan Fang et al. [26] for the older age group in Guangzhou. However, temperatures here are similar to that of the all-age experiments in Shanghai [27] and Changsha [20]. The reason for this difference is related to the age, physical fitness, and other characteristics of the subjects of this experiment. Young university students have better perception and adaptability to changes in the thermal environment and possess higher thermal tolerance [28], i.e., higher individual adaptive capacity. This finding proves that the age selection of the study subjects impacts the experimental results.

In terms of geographical space differences in experimental sites, the thermal nature temperature (range) and thermally acceptable temperature range are compared with relevant research results in the northern hemisphere in recent years in the study (Table 4). The comparison results reveal that the thermally neutral temperature in Hangzhou is lower than that in Guangzhou [3,27] and similar to that in Shanghai and Changsha [20]; the thermally neutral temperature range is similar to that in Changsha [20], Shanghai [29] and Qingdao [30] where there is a subtropical monsoon climate. The reason for the difference in thermal nature temperature between Hangzhou and Guangzhou [3] is that although both places are located in the same humid and hot region, Guangzhou is located at a lower latitude and has higher temperatures throughout the year. The result of a higher overall threshold of thermal nature temperature range is consistent with the characteristics of the Guangzhou population adapted to a warm and hot climate. Warsaw [31], located in a temperate continental climate zone at high latitudes, has the widest thermal nature temperature range. The latitude of Hangzhou is between that of Guangzhou and that of Warsaw. It is similar to that of Shanghai and Changsha. The distribution of perceptual thresholds for the thermal environment of the tested population is more moderate. It can be assumed that the higher the latitude of the city is or the more drastic the variation of urban meteorological factors is, the stronger the thermal adaptability of the population will be and the broader the thermal acceptance range will be. This hypothesis is also confirmed by the results of the comparison of the thermally acceptable temperature range. The thermally acceptable temperature range of Hangzhou is similar to that of Dhaka [32] in the tropical monsoon climate zone, Taichung [33] in the tropical region, and Xi’an [26] in the temperate monsoon climate zone. In summary, the thermal nature temperature, thermal nature temperature range, and thermally acceptable temperature range are correlated with latitude in addition to climate zones.

### 4.4. Empirical Model

In the study, meteorological indicators such as air temperature, solar radiation, mean wind speed, and relative air humidity are fitted to the MTSV results linearly. The analysis is conducted based on multiple linear regression. Thermal sensory models for all seasons of the year are developed. The significance *p*-value of all models is less than 0.01 (*p* < 0.01), and the value of variance inflation factor VIF, which measures the degree of multicollinearity, is less than 10 (VIF < 10), indicating that the regression models are statistically significant.

The formula for each season of the whole year is shown as follows:

MTSV (Transition season) = 0.151 *T_a_* −0.041 *V_a_* −3.268 (R^2^ = 0.818, *p* < 0.01).

MTSV (Summer) = 0.136 *T_a_* + 0.004 *SR*−0.018 *V_a_* −4.983 (R^2^ = 0.897, *p* < 0.01).

MTSV (Winter) = 0.150 *T_a_* + 0.003 *SR* − 0.010 *V_a_* −3.942 (R^2^ = 0.769, *p* < 0.01).

MTSV (Annual) = 0.113 *T_a_* + 0.002 *SR*−0.084 *V_a_* −2.242 (R^2^ = 0.810, *p* < 0.01).

In the formula: MTSV is the mean thermal sensation vote; *T_a_* is air temperature; *SR* is solar radiation; *RH* is the relative humidity of air; *V_a_* is mean wind speed.

The correlations established in the model perform well, and the Pearson correlation coefficients are close to each other. All are at a high level. This indicates that the actual climatic conditions in the West Lake scenic area of Hangzhou are in good agreement with the TSV voting results. The statistical results can be applied to the thermal sensory assessment of the area.

The models in all seasons indicate that air temperature and solar radiation indicators are positively correlated with MTSV, and the increase in air temperature and solar radiation values will lead to an increase in heat perception temperature. Among them, the coefficient of air temperature on heat perception is higher than that of solar radiation. The model also demonstrates that the average wind speed is negatively correlated with MTSV. The higher the average wind speed is, the lower the MTSV is. The result is highly consistent with the experimental findings in Yokohama [39], six European cities [11], and Hong Kong [40].

Each meteorological indicator may impact MTSV, but the influence coefficient of each meteorological indicator on thermal comfort is different in each season. The coefficient of influence of solar radiation on MTSV in the transitional season is 0, which can be neglected, so it can be said that the influence coefficients of the transitional season model exclude the coefficient of solar radiation. In general, air temperature has the most significant impact on thermal comfort, followed by average wind speed. Solar radiation has the weakest impact on thermal comfort. The comparison of the differences between the seasonal models shows that air temperature and average wind speed are the most important climate indicators impacting thermal comfort in the more comfortable weather conditions of the transitional season; At the same time, solar radiation also impacts human thermal perception under extreme climates of winter and summer.

In addition, there is a strong covariance between air humidity and mean wind speed indicators. The air relative humidity variable is automatically stripped during the regression analysis. This is different from the study conclusion of Ahmadi et al. [41]. The study result shows that temperature increases while precipitation decreases. However, the conclusion reached in the study is similar to that reached by Fang et al. [42] in Guangzhou. It’s proposed in the study variables-mean wind speed, or air relative humidity, is missed in empirical models for spring, summer, autumn, and winter. However, both variables are covered in empirical models in Hong Kong [40], Singapore [43], and Changsha [20]. This geographical difference once again proves that in the process of dissecting the constituent variables and their relationships, different assessment methods and models must be used for different regions and different research subjects and finding prediction methods adapted to local thermal conditions is an important task, which can improve the accuracy of thermal prediction.

## 5. Conclusions

In the study, the thermally neutral temperature and thermally acceptable temperature range of Hangzhou West Lake scenic area in a humid and hot region are discussed. Meteorological data from 18 days are collected in four typical landscape garden spaces around Hangzhou West Lake, and 360 valid questionnaires are gathered.

(1)It is found that the thermal evaluation based on the median method is more consistent with the actual thermal perception in Hangzhou West Lake. Although this study is limited to the youth group, it can still provide a reference basis for the design and research of climate adaptation in Hangzhou West Lake scenic area and urban outdoor spaces with similar climatic conditions.(2)The thermally neutral temperatures (ranges) and thermally acceptable ranges for the transitional season, summer, winter, and the whole year in Hangzhou West Lake scenic area are as follows thermal neutral temperature is 22.2 °C for the transitional season, 23.6 °C for summer, 12.0 °C for winter and 21.0 °C for the whole year. The thermal neutral temperature range is 10.3–22.6 °C for the transitional season, 18.4–24.35 °C for summer, and 10.2–21.5 °C for winter. The acceptable thermal temperatures range from 10.8–25.7 °C in the transitional season, 17.4–26.4 °C in the summer, 7.1–24.4 °C in the winter, and 13.0–25.7 °C throughout the year.(3)The interviewed group has the lowest acceptance of outdoor spatial climate conditions in winter, moderate acceptance in summer, and the greatest acceptance in the transitional season. Therefore, in terms of the climate-adapted spatial design carried out around the West Lake scenic area in Hangzhou, priority should be given to meeting the thermal demand in winter and summer.(4)Seasonal empirical models show that air temperature and average wind speed are the most important meteorological factors impacting the thermal evaluation of Hangzhou West Lake scenic area. In addition, solar radiation also plays an important role in extreme climate.

This study will have the same positive influence on West Lake scenic tourism planning. Moreover, it is of practical significance in the spatial design of climate adaptation. Based on the existing research, the team will carry out in-depth research on the direction of landscape architecture microclimate adaptive design, which is couples of times and spatial. The major direction will be focused on the tour route planning of microclimate adaptive in scenic tourist areas and the dynamic assessment of all-age thermal comfort of urban public green space.

Note: all pictures in the paper are drawn by the authors.

## Figures and Tables

**Figure 1 ijerph-19-14677-f001:**
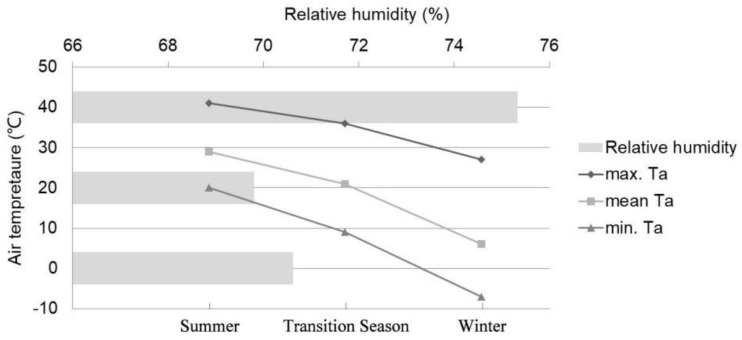
Overview of Hangzhou meteorological data in the past 30 years.

**Figure 2 ijerph-19-14677-f002:**
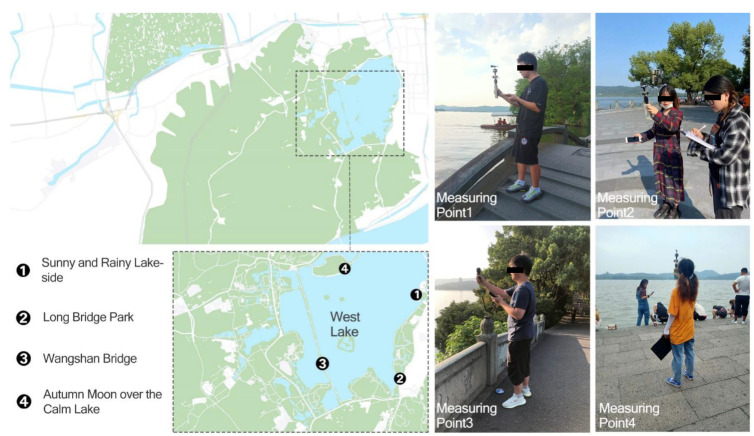
Location map of measurement points and experiment photos (repainting based on Baidu map).

**Figure 3 ijerph-19-14677-f003:**
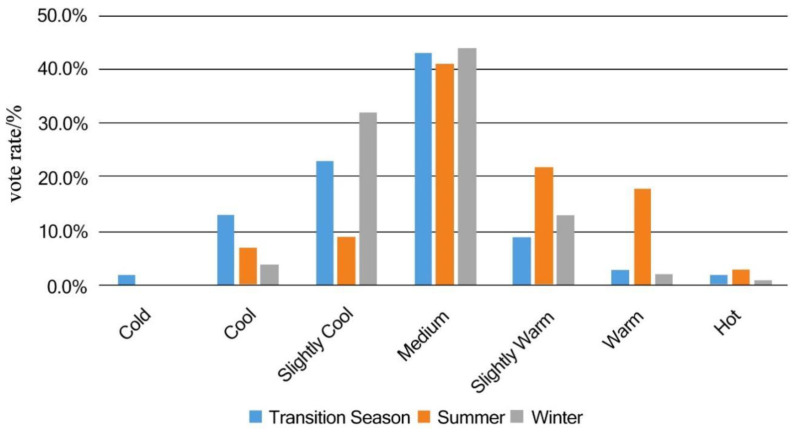
TSV seasonal comparison.

**Figure 4 ijerph-19-14677-f004:**
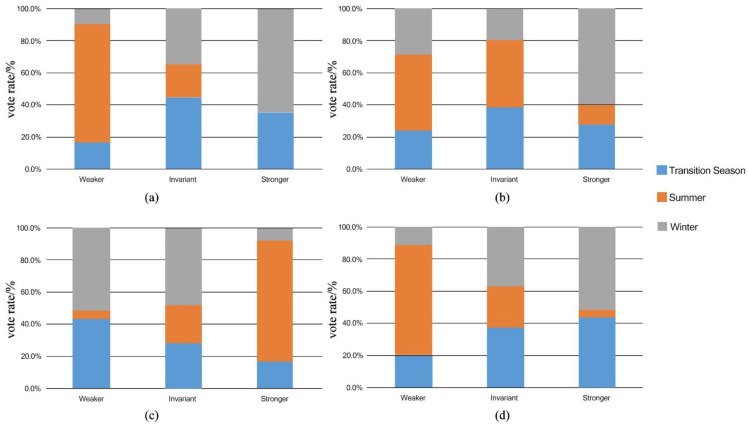
Preferential voting diagram for (**a**) air temperature, (**b**) solar radiation, (**c**) wind speed and (**d**) air humidity.

**Figure 5 ijerph-19-14677-f005:**
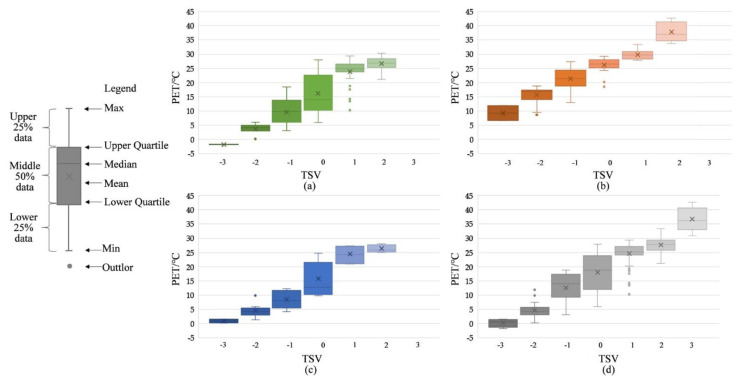
PET boxplots for (**a**) transitional season TSV, (**b**) summer TSV, (**c**) winter TSV and (**d**) year-round TSV.

**Figure 6 ijerph-19-14677-f006:**
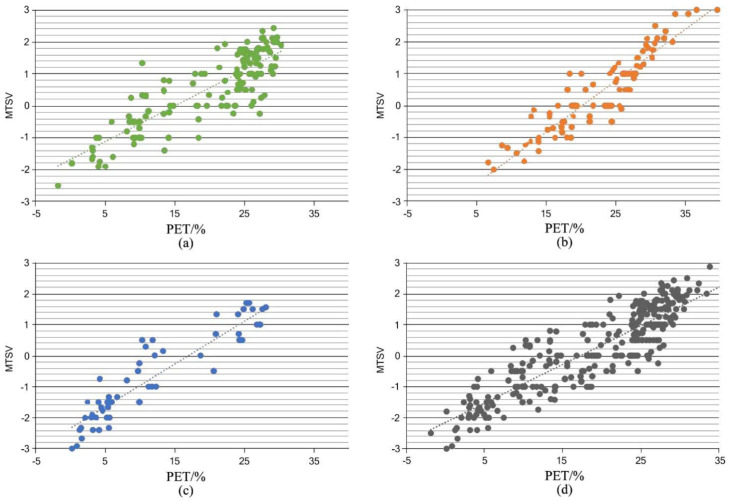
Linear regression analysis of PET and MTSV in (**a**) transitional season, (**b**) summer, (**c**) winter, and (**d**) year-round.

**Table 1 ijerph-19-14677-t001:** Actual measurement time.

	Transition Season	Summer	Winter
2020	22–24 September	16–18 July	22–24 January
2021	11–13 May 21–23 October	20–25 August	

**Table 2 ijerph-19-14677-t002:** statistics of meteorological test results.

Season	Parameter	Max	Mean	Min	SD
Transitional season	Solar (W/m^2^)	1073.0	401.5	10.0	411.1
	Ta (°C)	46.3	46.3	17.1	8.2
	RH (%)	65.4	48.4	41.9	6.4
	Va (m/s)	3.3	2.3	0.3	0.6
	PET (°C)	60.1	30.4	11.7	14.6
Summer	Solar (W/m^2^)	794.0	254.3	17.0	235.8
	Ta (°C)	43.6	32.3	25.8	5.3
	RH (%)	72.3	50.9	37.8	9.0
	Va (m/s)	2.2	0.9	0.0	0.5
	PET (°C)	57.9	34.1	22.3	10.6
Winter	Solar (W/m^2^)	558.0	263.7	0.0	214.4
	Ta (°C)	20.4	11.3	0.8	4.9
	RH (%)	71.0	32.4	15.2	12.7
	Va (m/s)	4.4	1.8	0.0	0.9
	PET (°C)	22.8	8.3	−4.7	7.4

**Table 3 ijerph-19-14677-t003:** Comparison of the thermal neutral temperatures and thermal neutral temperature ranges derived from the two methods.

	MTSV = 0 Method (°C)			Median Method (°C)		
Season	Thermal Neutral Temperature MTSV = 0	Thermal Neutral Temperature Range (−0.5 ≤ MTSV ≤ 0.5)	Thermal Acceptable Temperature Range (−1 ≤ MTSV ≤ 1)	Thermal Neutral Temperature (Median)	Thermal Neutral Temperature Range (Upper Quartile-Lower Quartile)	Thermal Acceptable Temperature Range > 90%
Transitional season	15.0	10.5–19.5	6.1–24.0	22.2	10.5–25.7	10.8–25.7
Summer	20.3	17.1–23.4	14.1–26.5	23.6	17.1–26.4	17.4–26.4
Winter	17.0	13.3–20.6	9.7–24.2	12.0	7.1–24.4	7.1–24.4
Annual	17.3	13.3–21.3	9.3–25.3	21.0	13.3–23.9	13.0–25.7

**Table 4 ijerph-19-14677-t004:** Comparison of results of outdoor thermal sensation studies.

City	Latitude, Longitude	Köppen Climate Zone	Thermal Neutral Temperature (°C)	Thermal Neutral Temperature Range (°C)	Thermal Acceptable Temperature Range (°C)
Changsha (China) [20]	28.26 N, 112.99 E	Cfa	14.9 (Winter) 23.3 (Summer)	15–22 (Annual)	21.1–29.2 (Annual)
Tehran [2]	35.78 N, 51.45 E	BSk		13.9–20.5 (Annual)	22.1–28(Annual)
Warsaw [31]	52.23 N, 21.12 E	Dfb	32.4 (Summer) –2.1 (Winter)	27.3–31.7 (Summer) 6.3–21.8 (Winter)	
Dhaka [32]	23.71 N, 90.41 W	Dfb	24.6 (Summer)		29.5–32.5 (Summer)
Taichung (China) [33]	24.13 N, 120.56 E	Cwa	25.6 (Summer) 23.7 (Winter)		21.3–35.9 (Annual)
Chandigarh [34]	30.73 N, 76.91 E	Cwa	29.5 (Summer)		20.2–36.3 (Annual)
			23.3 (Winter)		
Anatolia [11]	37.87 N, 32.49 E	BSk	26.8(Summer)		21.6–32.0 (Summer)
Rome [12]	41.90 N, 12.48 E	Csa			21.1–29.2 (Annual)
Melbourne [35]	37.81 S, 145.04 E	Cfb	20 (Summer)		20–24.9 (Summer)
			24.4 (Winter)		
Guangzhou (China) [3]	23.14 N, 113.28 E	Cfa	25.6 (Summer)	23.8–27.4(Summer)	22.6–31.2 (Summer)
Haikou (China) [36]	20.07 N, 110.34 E	Cfa	23.8 (Spring) 23.3 (Autumn) 19.2 (Winter)	21.4–27.1 (Spring) 19.2–32 (Autumn) >15.9 (Winter) 19.6–29.5 (Other three seasons)	
Shanghai (China) [37]	31.23 N, 121.49 E	Cfa		16.9–29.0 (Summer) 9.05–22.6 (Winter)	
Xi’an (China) [26]	34.34 N, 108.94 E	Cwa		12.4–26.9 (Annual)	9.8–30.7 (Annual)
Qingdao (China) [30]	36.08 N, 120.39 E	Cwa	13.7 (Winter)	6.3–21.0 (Winter)	
Mianyang (China) [38]	31.48 N, 104.67 E	Et	23.5 (Winter)		
			22.8 (Summer)		
Hangzhou (The study)	30.25 N, 120.16 E	Cfa	22.2 (Transition season) 23.6 (Summer) 12.0 (Winter) 21.0 (Annual)	10.5–25.7 (Transition season) 17.1–26.4 (Summer) 7.1–24.4 (Winter) 13.3–23.9 (Annual)	10.8–25.7 (Transition season) 17.4–26.4 (Summer) 7.1–24.4 (Winter) 13.0–25.7 (Annual)

## Data Availability

The data presented in this study are available on request from the corresponding authors.

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
