# Peer review of "Research on Seasonal Thermal Neutral Temperature in West Lake Scenic Area of Hangzhou, China"

_ijerph, 2022, doi:10.3390/ijerph192214677_

Round 1

Reviewer 1 Report

Dear Editors 

I have read a great paper, I only ask to review the references section and adjust them to MDPI standards and include in the conclusions section recommendations for future work. 

Regards 

edwin

Author Response

Reply: Thank you very much for your precise comments and suggestions, the manuscript has been thoroughly checked and carefully revised. The major revision has marked in red color.

Q1: English should be revised.

A1: The English grammar and writing have been checked and revised.

Q2: More scientific references are needed on the topic. Please see some of latest reputed

references on this topic.

A2: In order to improve the quality of the paper, the authors have added more 6 latest reputed references in the references section include the one you suggested.

Q3: The authors should clearly emphasize the novelty and originality of the paper compared with the available literature.

A3: The novelty and originality of the paper has been revised, as follow:

The literature on the thermal comfort of urban waterfront tourist attractions is still relatively rare. The existing inter-seasonal comparative studies mostly focus on the urban settlement environment. The research object groups are mostly local residents, and the thermal comfort is basically at a stable level. There are obvious differences between the scenic tourist areas and the urban settlements when talking about the thermal comfort perception. Hangzhou West Lake Scenic Area is an internationally famous tourist attraction, attracting a large number of domestic and foreign tourists. The upcoming Asian Games in 2023 will usher in more visitors to Hangzhou West Lake. Compared with local residents, these tourists from different regions are more likely to have significantly different physiological responses to thermal environment experience during a short trip. The social application value of this study is mainly reflected in that it can provide a clearer choice for tourists with different needs in different seasons to better enhance the tourism experience. The theoretical value is to compare the differences of thermal comfort in different seasons, explore the internal mechanism of seasonal variation of thermal comfort, and provide relevant basis for the later planning and design of Hangzhou West Lake Scenic Area and similar tourist attractions.

Q4: Fig. 6 may be deleted.

A4: Fig.6 helps to clearly demonstrate the correlation between PET and MSTV, and contribute to the rollout of the seasonal empirical models. Without it, readers may encounter confusion in derivation process, that’s the reason why we propose to retain it.

Transitional season: MTSV = 0.1118PET-1.6786 (R²=0.714, p < 0.01).

Summer: MTSV=0.1596PET-3.2371 (R²=0.842, p<0.01).

Winter: MTSV=0.1378PET-2.3387 (R²=0.723, p<0.01).

Annual: MTSV=0.1249PET-2.1588 (R²=0.754, p<0.01).

Figure 6 Linear regression analysis of PET and MTSV

Q5: Include thermal sensations detail.

A5: The conclusion of thermal sensation in this study was derived from the thermal sensation vote (TSV). Thermal sensation which closely related to the personal experience, can be influenced by environment but cannot be precisely measured or descried. Thus the thermal sensation of subjects could only be found in form of questionnaires with content of a seven-level index consistent, which is the ASHRAE thermal sensation scale (-3~+3). The result of questionnaire has shown in Figure 3.

The discussion of thermal sensation corresponds to the thermally neutral temperature and thermal acceptability in this paper. The final results of this paper show that the transition season has the largest number of voters for optimal thermal comfort, and the best thermal sensation.

Q6: Modify introduction section

A6: The introduction section has been revised follow:

The literature on the thermal comfort of urban waterfront tourist attractions is still relatively rare. The existing inter-seasonal comparative studies mostly focus on the urban settlement environment. The research object groups are mostly local residents, and the thermal comfort is basically at a stable level. There are obvious differences between the scenic tourist areas and the urban settlements when talking about the perception of thermal comfort. Hangzhou West Lake Scenic Area is an internationally famous tourist attraction, attracting a large number of domestic and foreign tourists. The upcoming Asian Games in 2023 will usher in more visitors to Hangzhou West Lake. Compared with local residents, these tourists from different regions are more likely to have significantly different physiological responses to the thermal experience during a short trip. The social application value of this study is mainly reflected in the fact that it can provide a clearer choice for tourists with different needs in different seasons to better enhance the tourism experience. That is why seasonal thermal neutral temperature research in West Lake should be conducted. In this study, the youth group of college students is selected as the research target. The following objectives will be achieved through the analysis of the measured data:

(1) to clarify the thermal evaluation method applicable to the West Lake of Hangzhou.

(2) to evaluate the seasonal differences in thermal perception of the recreational space of the West Lake of Hangzhou, and furthermore to derive the thermal nature temperature, thermal nature temperature range, and thermally acceptable temperature range by season.

(3) to examine the meteorological variables influencing the thermal perception of Hangzhou's West Lake and to suggest an empirical model for thermal perception prediction and its seasonal variations.

(4) to contribute to the design of landscape microclimate adaptability and crowd tour route.

Q7: Rephrase conclusion section 2

A7: The conclusion has been revised as follow:

In the study, the thermally neutral temperature and thermally acceptable temperature range of Hangzhou West Lake scenic area in a humid and hot region are discussed. Meteorological data of 18 days are collected in four typical landscape garden spaces around Hangzhou West Lake, and 360 valid questionnaires are gathered.

  • It’s found that the thermal evaluation based on the median method is more consistent with the actual thermal perception in Hangzhou West Lake. Although this study is limited to the youth group, it can still provide a reference basis for the design and research of climate adaptation in Hangzhou West Lake scenic area and urban outdoor spaces with similar climatic conditions.
  • The thermally neutral temperatures (ranges) and thermally acceptable ranges for the transitional season, summer, winter, and the whole year in Hangzhou West Lake scenic area are as followed: thermal neutral temperature is 22.2 ℃ for the transitional season, 23.6 ℃ for summer, 12.0 ℃ for winter and 21.0 ℃ for the whole year. Thermal neutral temperature range is 10.3-22.6 ℃ for the transitional season, 18.4-24.35 ℃ for summer, 10.2-21.5 ℃ for winter. The thermal acceptable temperatures range 8-25.7 ℃ in the transitional season, 17.4-26.4 ℃ in the summer, 7.1-24.4 ℃ in the winter, and 13.0-25.7 ℃ throughout the year.
  • The interviewed group has the lowest acceptance of outdoor spatial climate conditions in winter, moderate acceptance in summer, and the greatest acceptance in the transitional season. Therefore, in terms of the climate-adapted spatial design carried out around the West Lake scenic area in Hangzhou, priority should be given to meeting the thermal demand in winter and summer.
  • Seasonal empirical models show that air temperature and average wind speed are the most important meteorological factors impacting the thermal evaluation of Hangzhou West Lake scenic area, in addition solar radiation also plays an important role in the extreme climate.
  • This study will have same positive influence in West Lake scenic tourism planning. Moreover, it’s of practical significance of the spatial design of climate adaptation.

Q8: Improve quality of Fig. 5

A8: For facilitate comparison and presentation, the quality of data and marks were adjusted in manuscript.

Figure 5 PET boxplot

Q9: What is new in this study?

A9: The novelties of this study are:

  • Two common research methods of thermal comfort evaluation were compared, the median method was proved to be more suitable at the study plot. The study derived detailed thermal comfort temperature, thermal neutral temperature range, and thermal acceptable range for West Lake scenic area in Hangzhou.
  • After comparing the thermal comfort acceptability of each season, it is found that the thermal acceptability is ranked as transitional season > summer > winter, which leads to the conclusion that the thermal comfort design of landscape space in Hangzhou West Lake Scenic Area should meet the requirements of winter in a limited way, followed by summer, and finally consider the transitional season.

Thanks again, and wish you all the best!

Round 2

Reviewer 2 Report

The manuscript substaintially imroved  and authors have addreesed my all comments. It is recommended  for acceptance